# Resistance to Artemisinin Combination Therapies (ACTs): Do Not Forget the Partner Drug!

**DOI:** 10.3390/tropicalmed4010026

**Published:** 2019-02-01

**Authors:** Christian Nsanzabana

**Affiliations:** 1Department of Medicine, Swiss Tropical and Public Health Institute, CH-4051 Basel, Switzerland; christian.nsanzabana@swisstph.ch; Tel.: +41-61-284-8252; 2University of Basel, P.O. Box, CH-4003 Basel, Switzerland

**Keywords:** antimalarial, resistance, molecular marker, drug, surveillance, artemisinin, partner drug

## Abstract

Artemisinin-based combination therapies (ACTs) have become the mainstay for malaria treatment in almost all malaria endemic settings. Artemisinin derivatives are highly potent and fast acting antimalarials; but they have a short half-life and need to be combined with partner drugs with a longer half-life to clear the remaining parasites after a standard 3-day ACT regimen. When introduced, ACTs were highly efficacious and contributed to the steep decrease of malaria over the last decades. However, parasites with decreased susceptibility to artemisinins have emerged in the Greater Mekong Subregion (GMS), followed by ACTs’ failure, due to both decreased susceptibility to artemisinin and partner drug resistance. Therefore, there is an urgent need to strengthen and expand current resistance surveillance systems beyond the GMS to track the emergence or spread of artemisinin resistance. Great attention has been paid to the spread of artemisinin resistance over the last five years, since molecular markers of decreased susceptibility to artemisinin in the GMS have been discovered. However, resistance to partner drugs is critical, as ACTs can still be effective against parasites with decreased susceptibility to artemisinins, when the latter are combined with a highly efficacious partner drug. This review outlines the different mechanisms of resistance and molecular markers associated with resistance to partner drugs for the currently used ACTs. Strategies to improve surveillance and potential solutions to extend the useful therapeutic lifespan of the currently available malaria medicines are proposed.

## 1. Background

The emergence and spread of *Plasmodium falciparum* parasites with decreased susceptibility to artemisinin derivatives, and subsequent treatment failures after treatment with artemisinin-based combination therapies (ACTs) in the Greater Mekong Subregion (GMS) have raised concerns about the loss of the only highly-effective treatment currently available to treat malaria [1,2]. Artemisinin resistance, defined as delayed parasite clearance following treatment with conventional ACT regimens, has been associated with *pfKelch13* mutations in the GMS, and several mutations have been validated as molecular markers of artemisinin resistance in that region [3,4]. Artemisinin resistance may have spread or emerged in Eastern India, where delayed parasite clearance combined with a high rate of parasite survival in the ring survival assay (RSA) and the presence of certain *pfKelch13* mutations have been observed [5]. Moreover, some *pfKelch13* mutations associated with delayed parasite clearance have been also reported in South America [6,7] and Papua New Guinea [8]. Nevertheless, different *pfKelch13* mutations have been observed in Africa, but these were not associated with artemisinin resistance [9,10,11]. The mode of action of artemisinins is still not fully understood, even though artemisinin derivatives have been shown to cause oxidative stress and are probably impacting on multiple targets in the parasite, definitely implying a more complex resistance mechanism associated with different determinants other than *pfKelch13* [12,13]. Therefore, there is a possibility that parasites with decreased artemisinin susceptibility are already present in some places in Africa, and are circulating at low levels, due to the still high multiplicity of infections in this region, and the related semi-immunity in older individuals [14]. Moreover, partner drugs do play a role in parasite clearance, even though to a lesser extent than artemisinin derivatives, but their decreased efficacy is probably affecting parasite clearance, as well [15]. In fact, even though artemisinin derivatives are highly potent antimalarials that can reduce parasite biomass very quickly [16]; due to their short half-life, they need to be combined with partner drugs with long half-life to clear the remaining parasites after a 3-day ACT treatment course [17]. Indeed, ACT failure is not only due to artemisinin resistance, but also to the failure of partner drugs [18,19]. There are currently five ACTs recommended by the World Health Organization (WHO): artesunate-sulfadoxine-pyrimethamine (ASSP), artemether-lumefantrine (AL), dihydroartemisinin-piperaquine (DP), artesunate-amodiaquine (ASAQ), and artesunate-mefloquine (ASMQ) [20]. Artesunate-pyronaridine (ASPY) is now listed under the WHO’s Model List of Essential Medicines, and previous restrictions due to concerns about its hepatotoxicity effects have been removed [21,22]. In this review, the development of resistance to ACTs is discussed and strategies to improve antimalarial drug resistance surveillance systems that should pay the same attention to partner drug resistance are proposed, as partner drug resistance may exist or occur even before resistance to artemisinins emerges in high endemic settings.

## 2. Mechanisms of Resistance to Partner Drugs

### 2.1. Lumefantrine

Artemether-lumefantrine (AL) is the most widely used antimalarial in endemic countries. In 2017, it was estimated that it accounted for almost 75% of all procured quality-assured ACTs [23]. The mode of action of lumefantrine is not well understood, and its main target remains unresolved, even though it is thought to interfere with hem detoxification [24], or to directly inhibit *pfmdr1* [25]. Some recent reports have shown a decreased efficacy of AL in Angola [26,27], Gambia and Malawi [28]; however, so far no convincing evidence of lumefantrine resistance has been reported from the field [29]. Moreover, it has been shown recently in Angola that efficacy was higher when the full treatment course was directly observed [30]. Decreased susceptibility to lumefantrine (LUM) has been associated with gene copy number variations (CNV) in *pfmdr1* [31] and the wild type *pfcrt* (K76) and *pfmdr1* (N86) that confer increased susceptibility to chloroquine [32,33,34]. However, large meta-analyses failed to find a correlation between *pfmdr1* CNV and treatment failure, but confirmed the association with of *pfcrt* K76 and *pfmdr-1* N86 polymorphisms [35,36]. Moreover, laboratory selection of LUM resistant parasites has shown that multiple protein transporters may be involved in LUM resistance [37], and other investigations have shown that *pfmrp1* was associated with decreased susceptibility to LUM in vitro [38]. 

### 2.2. Amodiaquine

The combination of artesunate and amodiaquine (ASAQ) is the second most used ACT in malaria endemic settings. In 2017, it was estimated to account for >20 % of all good quality delivered ACTs [23]. Furthermore, amodiaquine (AQ) is used in combination with sulfadoxine-pyrimethamine (SP) for seasonal malaria chemoprevention in the Sahel region [39]. The mode of action of AQ is similar to chloroquine; the drug accumulates in the digestive vacuole where it binds the toxic hem, preventing its formation into the inert form hemozoin [40]. AQ resistance has been associated with point mutations in the *pfcrt* and *pfmdr1* genes, and the same mutations as those for chloroquine resistance (*pfcrt* 76T and *pfmdr1* 86Y) have been shown to be the main determinant of decrease susceptibility to AQ in vitro and in vivo [34,41,42]. However, a specific *pfcrt* haplotype of codons 72 to 76 (SVMNT) has been associated with resistance to AQ, but to a lesser degree of resistance to chloroquine [43,44,45].

### 2.3. Piperaquine

The combination of dihydroartemisinin and piperaquine (DP) has proven to be an efficacious treatment in most malaria endemic settings [46], and is used for treatment in some countries in Southeast Asia and Africa [47]. Dihydroartemisinin and Piperaquine are also increasingly considered for malaria prevention in pregnancy [48,49], and for mass drug administration (MDA) in near-elimination settings in Africa [50,51,52]. However, DP is failing in the sub-Mekong region due to resistance to piperaquine (PQ) and decreased susceptibility to artemisinin derivatives [53,54]. Treatment failure with DP was first observed in Western Cambodia, where the cumulative risk of treatment failure increased from 9.2 % to 24.1% between 2008 and 2010 [18]. The mode of action of PQ was initially linked to the inhibition of one or more steps in the hemoglobin degradation pathway, but it is not yet fully understood [55,56]. More recently, PQ resistance has been associated with CNV in *plasmepsin II and III* [57] and point mutations in *pfcrt* [56,58,59]. As the classic in vivo assays have shown limitations in the assessment of phonotypical resistance to PQ, a piperaquine survival assay (PSA) has been established to assess changes in parasite strains’ susceptibility to PQ [57]. The presence of different parasite populations with different mechanisms of resistance to PQ is intriguing and warrants further investigations to elucidate this pleiotropic mechanism of resistance [59].

### 2.4. Sulfadoxine-Pyrimethamine

Despite widespread resistance to sulfadoxine-pyrimethamine (SP) in most malaria endemic settings, the combination artesunate and SP (ASSP) is still used in a few countries for malaria treatment [47] and intermittent preventive treatment in pregnancy (IPTp) [49], and in combination with amodiaquine for seasonal malaria chemoprevention (SMC) [39]. Sulfadoxine and pyrimethamine are inhibiting two enzymes involved the folate biosynthesis pathway; dihydropteroate synthase DHPS and dihydrofolate reductase DHFR, respectively [60,61]. Resistance to sulfadoxine and pyrimethamine is by far the best characterized mechanism of resistance; it is associated with point mutations in the *pfdhps* and *pfdhfr* genes, respectively [62,63]. Indeed, the accumulation of point mutations in both genes is associated with increasing levels of resistance to the combination of the two drugs, with the combination of the triple mutant in *pfdhfr* (51I, 59R, 108N) and the double mutant in *pfdhps* (437G and 540E) being associated with an increased risk of in vivo treatment failure. This quintuple mutant has been shown to have emerged in Southeast Asia before it spread to other malaria endemic areas, notably Africa [64,65]. The presence of the quintuple mutation is used to guide treatment policy for intermittent preventive treatment in infants (IPTi): WHO is recommending not to use IPTi in areas where the *pfdhps* 540E mutation (a surrogate marker for the presence of the *pfdhfr*/*pfdhps* quintuple mutant) does exceed 50% [66].

### 2.5. Mefloquine

The combination of artesunate and mefloquine (ASMQ) was the first ACT that has been introduced in Southeast Asia to stop resistance to mefloquine (MQ) monotherapy in the early 1990s [67]. Since, the use of this combination has historically been restricted to Southeast Asia, the Pacific region and South America, but was not used in Africa. With increasing levels of resistance to MQ and decreased susceptibility to artemisinin derivatives, the combination is now used only in a few countries [47]. The mode of action of MQ is still unclear, but the drug may inhibit hem detoxification [24,68], or directly inhibit *pfmdr1* [69]. Mefloquine resistance has been associated with CNV in *pfmdr1* [70,71], and polymorphisms in *pfmrp1* and *pfmrp2* could also potentiate MQ resistance [38,72,73]. More recently, the cytoplasmic ribosome (pf80S) of the asexual blood-stage parasite has been suggested as the main target of MQ [74]. 

### 2.6. Pyronaridine

The combination of artesunate and pyronaridine (ASPY) is not yet recommended by the WHO, but has received a positive opinion from the European Medical Agency (EMA) under article 58 [21]. Moreover, the product was added to the WHO’s Model List of Essential Medicines (EML) and Model List of Essential Medicines for Children (EMLc) in 2017, and is the only ACT indicated for the blood stage treatment of both *P. falciparum* and *P. vivax* [22]. The combination has shown high efficacy in Africa where its efficacy was non-inferior to AL and DP [75,76], and in Eastern Cambodia [77]. However, the efficacy of the combination has been shown to be low in Western Cambodia, probably due to artemisinin resistance and potential cross-resistance between pyronaridine (PY) and PQ [78]. The exact mode of action of PY is not well known; however, the drug is thought to interfere with the hemozoin formation [79]. To date, resistance to PY has not been reported, but ex vivo assessment has shown an association between decreased susceptibility to PY and the 76T mutation in *pfcrt* [80].

## 3. Discussion

The emergence and spread of parasites with decreased susceptibility to artemisinins is a major threat to malaria control and elimination. Monitoring antimalarial drug efficacy and resistance is of paramount importance to maintain the gains made over the last decades in reducing malaria burden and mortality. Surveillance systems using molecular markers of resistance should become standard practice, as they are easy to implement and can offer more updated information to complement the often sparse and outdated data from therapeutic efficacy studies. Indeed, molecular markers can provide useful information to policymakers, allowing them anticipating treatment efficacy changes over time, and eventually decide on treatment policy change before resistance translates to clinical failures. However, molecular surveillance should not only focus on artemisinin resistance, but also on partner drug resistance. 

Our understanding of the mode of action and mechanisms of resistance, even though still incomplete, has substantially improved, and validated molecular markers can be used to track the emergence and spread of resistance to antimalarial drugs, as well as to predict clinical efficacy or failure. For instance, CQ withdrawal in most of Africa lead to the re-emergence of CQ sensitive parasites with the expansion of wild-type *pfcrt* K76 that ultimately resulted in CQ efficacy restoration [81,82]. The fact that the two major ACTs in Africa (i.e., ASAQ and AL) were selecting in opposite directions on the *pfmdr1* and *pfcrt* genes [35,83] resulted in increasing efficacy of ASAQ in areas of high AL usage, where the efficacy of the latter was declining [34]. However, mechanisms of resistance may differ from one area to another: Whilst the restoration of CQ efficacy was associated with the return of the wild-type *pfcrt* K76 in Africa [82], restoration of CQ efficacy in French Guiana was associated with a new mutation in *pfcrt* (350R), with the 76T mutation remaining fixed in the parasite population [84]. Likewise, resistance to PQ has been associated with an elevated copy number of the *plasmepsin II* and *III* gens [54,57]; however, other studies have shown that PQ resistance was mediated through single nucleotide polymorphisms (SNPs) in *pfcrt* [56,58,59] making the molecular surveillance of PQ resistance more complex. Therefore, surveillance systems should not only aim at detecting known and validated molecular markers, but also at tracking any new genotypes that could be associated with antimalarial drug resistance. Whole genome sequencing (WGS) could be used in routine surveillance through regional reference laboratories with respective capacities [85,86] for defining potential new markers that, in turn, would be validated by in vitro phenotypic assays and/or gene editing techniques.

In high transmission settings, parasites are more exposed to sub-therapeutic drug concentrations of the long half-life partner drugs, thus increasing the chances of developing resistance [87]. The introduction of aggressive chemoprevention methods such as seasonal malaria chemoprevention (SMC) and mass drug administration (MDA) will put even more pressure on partner drugs [88,89]. The presence of parasites with decreased susceptibility to artemisinin derivatives does not necessarily lead to ACT treatment failure, as evidenced with a recent report from Myanmar, where despite high prevalence of *pfKelch13* mutations associated with delayed parasite clearance, AL retains its high clinical efficacy [90]. However, resistance to the partner drug would immediately lead to ACT failure, as with the current 3-day ACT regimen, the short half-life of artemisinin derivatives could not sustain the efficacy of the combination on its own [91,92]. Hence, molecular surveillance in high transmission settings should aim at using more sensitive techniques such as amplicon deep sequencing that can also detect minority variants potentially present in the parasite population [93,94,95], this could help to earlier predict the emergence and spread of resistant parasites, allowing policymakers to develop alternative treatment strategies, before resistance translates into clinical treatment failures.

Currently, there is no alternative to ACTs, even though the antimalarial drug pipeline is promising [96,97]. Strategies are needed to prolong the useful therapeutic lifespan of the current malaria medicines, including 1) extending the duration of the current 3-day regimen of ACTs [90,98,99]; 2) increasing the dose of the partner drugs [100,101]; 3) using triple combination therapies, with two partner drugs selecting in opposite directions [36,102,103]; and 4) utilizing multiple first-line treatments [36]. However, those may be solutions to preserve the efficacy of ACTs in the short- or medium-term only, as there is already some evidence of parasites developing resistance to two partner drugs and the artemisinin component at the same time [104,105]. The development of nano-based drug formulations is another strategy to fight drug resistance by improving drug targeting and dosing [106,107,108]. More resources should also be allocated to study the mode of action and mechanisms of resistance to new antimalarial drugs. Not only will this allow the discovery of new molecular markers for resistance surveillance, but also pave the way for the development of new drugs with different modes of action. 

The emergence and spread of artemisinin resistance are a major threat to the current efforts to control and eliminate malaria. Molecular markers are valuable tools for monitoring antimalarial drug resistance, and need to be fully integrated in routine surveillance, especially for partner drugs in high endemic settings, where resistance to partner drugs may emerge before resistance to artemisinin.

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
