# Peer review of "Resistance to Artemisinin Combination Therapies (ACTs): Do Not Forget the Partner Drug!"

_tropicalmed, 2019, doi:10.3390/tropicalmed4010026_

Round 1
Reviewer 1 Report
This paper presents a useful and detailed review of the current state of malaria parasite resistance to ACT's which will be of interest to readers.
A number of suggested amendments (mainly relating to English) are listed below; please note that this list does not include very minor corrections.
line 39; artemisinin causing should be artemisinin causes
line 40; implies definately should be definately implying
line 66, suggest changing later for recently
line 70, delete of
line 81, mutations than should be mutations as
line 88f, the phrase 'and is used.......' has been repeated twice
line 89, the phrase 'DP is also increasingly.......' has been repeated
line 96, 'and point mutations.....' is repeated
line 97, suggest that 'its limits to assess' to limitations in the assessment of'
lines 103-104, please clarify this seems to be saying that SP is used for IPT in pregnancy because of the high prevalance of SP resistance?
line 126, 'could potentiate as well' should be 'could also potentiate'
line 134, Al should be AL?
line 139, please clarify what is meant by 'mutations in the 76T mutation'
line 161, reference cited to Pelleau et al 2015 should be given a number
line 197, 'threat the current' should be 'threat to the current'
Author Response
Responses to reviewer 1 attached

Reviewer 2 Report
This is a review on ACT resistance that emphasizes the need to monitor resistance to drug partners of artemisinins. This comprehensive review provides an up-to-date information on artemisinin/ACT-resistance and should be very useful for general readers as well as for malaria specialists.
Major comments:
Lines 144-148: Please state clearly whether the author is advocating the use of molecular markers to guide antimalarial drug policy, rather than basing the policy decision on therapeutic efficacy studies. Which one is the gold standard to guide drug policy?
Minor comments:
Line 27, “The emergence and spread… HAVE raised concerns…” (please see line 197, “the emergence and spread of artemisinin resistance are…”)
Line 36: Moreover, (add a comma here)
Lines 39-41: Please check the sentence structure.
Line 46: “artemisinin are” – please correct
Line 52, ref 20: Please provide the complete bibliographic information: third edition, city, etc.
Lines 52-53, “Artesunate-pyronaridine previous restrictions…” Please rewrite this part of the sentence.
Line 60: A period (.) after “countries”
Line 62: even though (?) it is thought…
Line 81: same mutations as those for
Lines 88-89: Please delete “and is used for treatment in some countries in Southeast Asia and Africa” repeated three times.
Lines 90-91: Please delete “DP is also increasingly considered for malaria prevention in pregnancy” repeated twice.
Lines 102-105, “…because of the high prevalence of SP resistance, SP is still widely used for IPTp…” This part of the sentence sounds contradictory. The author should mean “in spite of” rather than “because.”
Lines 105-107: sulfadoxine and pyrimethamine inhibit dhps and dhfr, respectively. Please change the order (dhps and dhfr, and not dhfr and dhps).
Lines 113-116: Does the author mean that WHO does not recommend the use of SP in areas where 540E exceeds (or does not exceed, as stated in the manuscript) 50%? Please clarify.
Line 134: AL (not “Al”)
Lines 137 and 80: haemozoin (British spelling)
Line 138: “assessment have” – has
Line 139: …PY and 76T mutation in Pfcrt
Line 153: …the re-emergence (small letter “t” in “the”)
Line 161: Ref Pelleau et al. 2015 does not seem to be in the list of references. Ref 84 (Humphreys et al 2007) is in the reference list but was not cited in the text (?). Please re-check the reference citations.
Line 174: does not necessarily lead
Lines 186-187, ref 99-101: These 3 references are on mefloquine and seem to be old (1992. 1993, 2000). I am not sure if these references support the author’s statement. Please re-check.
Ref 40: Please check the journal name.
Ref 49 and 50: The journal name is Clin Infect Dis. Please complete these two references (volume, pages).
Ref 65, 70, 77, and 92: The journal name is Lancet.
Ref 81: Please check the journal name (is it repeated twice?).
Ref 95: Please provide the complete reference: volume, pages.
Ref 103: Please provide the complete reference: volume, pages.
Ref 105: Please provide more information on this reference: journal name? book title?
Ref 106: The journal name is Eur J Pharm Sci.
Author Response
Responses to reviewer 2 attached
